# A New Method for the Assessment of Myalgia in Interstitial Lung Disease: Association with Positivity for Myositis-Specific and Myositis-Associated Antibodies

**DOI:** 10.3390/diagnostics12051139

**Published:** 2022-05-04

**Authors:** Gianluca Sambataro, Chiara Alfia Ferrara, Carla Spadaro, Sebastiano Emanuele Torrisi, Giovanna Vignigni, Ada Vancheri, Giuseppe Muscato, Nicoletta Del Papa, Michele Colaci, Lorenzo Malatino, Stefano Palmucci, Lorenzo Cavagna, Giovanni Zanframundo, Francesco Ferro, Chiara Baldini, Domenico Sambataro, Carlo Vancheri

**Affiliations:** 1Department of Clinical and Experimental Medicine, Regional Referral Centre for Rare Lung Disease, A.O.U. “Policlinico-San Marco”, University of Catania, 95123 Catania, Italy; chiara.ferrara@virgilio.it (C.A.F.); carlaspadaro@live.it (C.S.); torrisiseby@hotmail.it (S.E.T.); giovannavignigni@gmail.com (G.V.); adact1@hotmail.it (A.V.); gpp.muscato@gmail.com (G.M.); vancheri@unict.it (C.V.); 2Outpatient Clinic of Rheumatology, Artroreuma S.R.L., Corso S. Vito 53, 95030 Mascalucia (CT), Italy; d.sambataro@hotmail.it; 3Day Hospital of Rheumatology, Department of Rheumatology, ASST G.Pini-CTO, Piazza Cardinal Ferrari 1, 20122 Milan, Italy; nicoletta.delpapa@asst-pini-cto.it; 4Internal Medicine Unit, Department of Clinical and Experimental Medicine, Cannizzaro Hospital, University of Catania, Via Messina 829, 95100 Catania, Italy; michele.colaci@unict.it (M.C.); malatino@unict.it (L.M.); 5Department of Medical, Surgical Sciences and Advanced Technologies, “G.F. Ingrassia”, University of Catania, Via S. Sofia 68 Catania, 95123 Catania, Italy; spalmucci@unict.it; 6Division of Rheumatology, University and IRCCS Policlinico S. Matteo, Piazzale C. Golgi 19, 27100 Pavia, Italy; lorenzo.cavagna@unipv.it (L.C.); gio.zanframundo@gmail.com (G.Z.); 7Department of Clinical and Experimental Medicine, Rheumatology Unit, University of Pisa, Via Roma 24, 56126 Pisa, Italy; francescoferrodoc@gmail.com (F.F.); chiara.baldini74@gmail.com (C.B.)

**Keywords:** myalgia, myositis, interstitial lung disease, tender points, autoantibodies, connective tissue disease, diagnosis, fibromyalgia

## Abstract

In this study, it was found that myositis-specific and myositis-associated antibodies (MSAs and MAAs) improved the recognition of idiopathic inflammatory myopathies (IIMs) in interstitial lung disease (ILD) patients. The objective of this study is to propose a clinical method to evaluate myalgia in respiratory settings as a possible tool for the recognition of MSA/MAA positivity in ILD patients. We prospectively enrolled 167 ILD patients with suspected myositis, of which 63 had myalgia evoked at specific points (M+ILD+). We also enrolled in a 174 patients with only myalgia (M+ILD-) in a rheumatological setting. The patients were assessed jointly by rheumatologists and pulmonologists and were tested for autoantibodies. M+ILD+ patients were positive for at least one MAA/MSA in 68.3% of cases, as were M-ILD+ patients in 48.1% of cases and M+ILD- patients in 17.2% of cases (*p* = 0.01 and <0.0001, respectively). A diagnosis of IIM was made in 39.7% of M+ILD+ patients and in 23.1% of the M-ILD+ group (*p* = 0.02). Myalgia was significantly associated with positivity for MSA/MAAs in ILD patients (*p* = 0.01, X^2^: 6.47). In conclusion, myalgia in ILD patients with suspected myositis is associated with MSA/MAA positivity, and could support a diagnosis of IIM. A significant proportion of M+ILD- patients also had MSA/MAA positivity, a phenomenon warranting further study to evaluate its clinical meaning.

## 1. Introduction

Interstitial lung diseases (ILDs) are characterized by an accumulation of extracellular matrix and cells (mainly fibroblasts and/or immune cells) in the lung interstitium, leading to tissue fibrosis and potentially progressive respiratory failure. A number of conditions are associated with ILD, of which autoimmune diseases represent at least 18% [1]. Among these, idiopathic inflammatory myopathies (IIMs) represent about 6% of total ILDs, affecting 65% of IIM patients and leading to death in 80% of cases [2]. IIMs can be associated with specific skin signs: Raynaud’s phenomenon (RP) or the classic triad of antisynthetase syndrome (AS) composed of ILD, arthritis and myositis. Nevertheless, the number of ILD-IIMs patients may be underestimated, because a significant proportion of these patients show only ILD at the diagnosis, with no or mild muscle involvement [3]. They tend to be referred to pulmonologists without a systematic rheumatological assessment, making the diagnosis very difficult to achieve, at least in some cases. Recently, the search for myositis-specific and myositis-associated antibodies (MSA/MAAs) has provided considerable help in diagnosing these diseases, considering that IIMs are frequently seropositive for at least one of these autoantibodies [4]. However, with the exception of Jo1, the majority of MSAs and MAAs are not included in the common anti-extractable nuclear antigen antibodies (ENA) panel, and the specific MSA/MAA panel is not yet widely diffused in clinical practice.

ILD-IIM patients commonly experience no or mild myalgia, and when present it is often associated with autoantibody positivity, but not with increased levels of muscle enzymes, such as creatine phosphokinase (CPK), aspartate transaminase (AST), lactic dehydrogenase (LDH) or myoglobin [5]. A possible explanation of this phenomenon was given by Noda K et al. [6]. In their interesting study, they reported an association of myalgia mainly with dermatomyositis (DM) patients, describing an inflammatory involvement limited to the muscular fascia with normal or slightly elevated muscle enzymes. In contrast, myalgia is less common in polymyositis (PM), despite inflammation of the muscle fibers with significant elevation of serum muscular enzymes.

Myalgia is an umbrella term that refers to muscular pain frequently reported in patients with various conditions not necessarily of a rheumatic nature. For this reason, it is included neither in the specific classification criteria for IIMs nor in the interstitial pneumonia with autoimmune features (IPAF) criteria [7,8,9]. However, in the respiratory setting, the clinical assessment of myalgia in the muscular fascia could be useful for diagnostic purposes, considering that ILD is more often associated with DM, AS or overlap conditions, whereas PM is associated with ILD in less than 10% of cases [10].

The objective of this study is to clinically evaluate myalgia in muscular fascia as a possible predictor of positivity for MSA/MAAs in ILD patients suspected of having an underlying IIM, enrolled in a respiratory setting.

## 2. Materials and Methods

This was a prospective observational, cross-sectional study conducted from August 2021 to March 2022. Inclusion criteria for the study were a new diagnosis of ILD and a suspicion of underlying IIM (defined as reported in clinical evaluation) with and without myalgia (the study group and first control group, respectively), and the presence of myalgia not associated with known ILD or systemic autoimmune conditions (the second control group). Patients were excluded in the absence of written informed consent or if they did not complete the diagnostic assessment.

Patients were enrolled consecutively in the Regional Referral Center for Interstitial and Rare Lung Diseases of the University of Catania and in the Artroreuma outpatient rheumatology clinic.

### 2.1. Patients

Patients with a new ILD diagnosis were clinically evaluated at the same time by pulmonologists and rheumatologists, looking for respiratory symptoms (e.g., dyspnoea and cough) and clinical signs suggestive of autoimmune conditions, such as RP, sclerodactyly, and telangiectasias, or skin signs suggestive of IIM (e.g., Gottron’s sign, Mechanic’s hands, or heliotrope rash).

ILD patients were enrolled if a underlying IIM was suspected. Based on the current validated criteria [9], IIM can be suspected when onset age ≥ 18 years, and in the presence of proximal weakness, typical skin rashes (Gottron’s sign/papules or heliotrope rash), increased levels of creatine phosphokinase (CPK), lactic dehydrogenase (LDH), aspartate and alanine transferase (AST and ALT), or dysphagia. However, we considered that serum levels of LDH are aspecific and often elevated in lung damage, and dysphagia is very common in all ILDs, particularly idiopathic pulmonary fibrosis (IPF) [11,12]. Despite not being included in any criteria for IIM, DM, or PM, we considered patients to be suspected of an underlying IIM also in the presence of any of the following features: inflammatory arthritis/polyarticular morning joint stiffness associated with increased erythrosedimentation rate (ESR) and/or C reactive protein (CRP) [6,9,10]; increased serum level of myoglobin or aldolase (at least 1.5 times the upper limit); antinuclear antibody (ANA) positivity with nucleolar or cytoplasmic pattern; nailfold videocapillaroscopy (NVC) positivity in patients seronegative for systemic sclerosis-associated antibodies; presence of bushy capillaries in NVC; presence of other suggestive skin rashes, such as mechanic’s hands and hiker’s feet; features of organizing pneumonia (OP) in high-resolution computed tomography (HRCT), or a history of cancer [13,14,15,16].

### 2.2. Main Outcome Variable

Myalgia was evaluated in the proximal muscles, which are generally affected in IIM [17]. In detail, myalgia was assessed by applying a weight of 4 kg to specific points in proximity to superficial entheses, in other words, in those parts in which the fascia is most represented. The points evaluated were the following: suboccipital muscle insertion, lesser tubercle of the humerus (point of insertion of the subscapularis muscle), median and lateral epicondyle (point of insertion of the flexor and extensor muscles, respectively), latero-posterior surface of sacrum (point of insertion of the gluteus maximus muscle), greater trochanter (point of insertion of the gluteus medium muscle), vastus medialis, and 2 cm above the patella. The points are graphically reported in Figure 1.

Myalgia was considered present (M+) when associated with a defensive retraction by the patient. Looking for oligo-amyopathic IIMs, we considered patients with more than 1/3 positive points (at least 5) to be M+.

The same rheumatologists who evaluated ILD patients also enrolled myalgia patients in a rheumatological setting as a control group. Myalgia in these patients was evaluated in the same manner as reported above. All of these patients were also questioned regarding symptoms or signs of autoimmune diseases, and did not display any feature suggesting an inflammatory condition.

### 2.3. Procedures

#### 2.3.1. Laboratory Assessment

All patients underwent the following laboratory tests: complete blood count, ESR, CRP, urine test, AST, ALT, myoglobin, aldolase, ferritin, CPK, LDH, complement fractions C3 and C4, serum protein electrophoresis, ANA, rheumatoid factor, anti-citrullinated protein antibodies (ACPA), classic ENA profile (anti-Ro60kD, anti-La, anti-Sm, anti-RNP, anti-Scl70, anti-Jo1), anti-neutrophil cytoplasmic antibodies (ANCA), and MSA/MAA.

ANAs were evaluated by indirect immunofluorescence and were considered positive when the titer was ≥1/80. ENA and MSA/MAAs were evaluated by immunoblotting (IB). MSA/MAAs were tested with the commercial Immunoblot Euroline Myositis Profile (Euroimmun, Lübeck, Germany) which evaluated 11 MSAs (anti-Mi2, Tif1γ, MDA5, NXP2, SAE1, SRP, PL7, PL12, Jo1, EJ, OJ) and 4 MAA (Pm/Scl, RNP, Ku, Ro52k) [18].

Laboratory assessment was performed by physicians independent of this study, within three months after the clinical evaluation.

#### 2.3.2. High-Resolution Computed Tomography

All ILD patients underwent HRCT at baseline. The presence of ILD was confirmed by HRCT with slices ranging from 1.25 mm to 2.5 mm at the baseline. Images were evaluated by expert clinicians trained in the evaluation of ILD patterns (radiologists and pulmonologists) through the presence of ground glass opacities, reticulations, honeycombing or nodules, according to specific interpretation guidelines [19].

To exclude the presence of unrecognized lung involvement, HRCT was also performed on M+ patients positive for MSAs.

#### 2.3.3. Complementary Exams

NVC was performing using VideoCap 3.0 (DsMedica, Milan, Italy Viale Monza 133, 95125). Two expert rheumatologists collected 3 images from the second to the fifth finger of both hands. The classification was made semi-quantitatively, according to Cutolo’s Criteria [20], but the presence of bushy capillaries was also recorded and highlighted [21]. NVC was performed on all ILD patients, considering that NVC in AS patients is not associated with the presence of RP [22].

Other assessments, such as electromyography, magnetic resonance of the muscles, Schirmer’s test or unstimulated whole saliva flow rate, minor salivary gland, lung, and muscle biopsy were performed when deemed clinically useful.

### 2.4. Diagnosis of Autoimmune Conditions

The diagnosis of autoimmune conditions was made according to the validated criteria endorsed by the American College of Rheumatology, European League Against Rheumatism, European Respiratory Society, and American Thoracic Society, updated to the latest version [7,9,23,24,25,26,27,28,29,30,31,32]. The only condition without validated criteria is currently AS. In this case, we used the Connors criteria [8]. The classification of undifferentiated connective tissue disease (UCTD) for ILD patients was made according to the criteria for IPAF [7].

The diagnosis of ILD patients was made by means of multidisciplinary team discussion, considered the gold standard for the assessment of these patients [33].

### 2.5. Statistical Analysis

Statistical analysis was performed with IBM SPSS Statistics for Windows, Version 20.0 (Armonk, NY, USA: IBM Corp.). For the evaluation of sample size, we considered a confidence interval of 5 and a confidence level of 95%. As already reported, IIMs accounts for about 6% of total ILD [2]. As our center has around 200 new ILD diagnoses each year, we considered the enrollment of at least 72 patients. We employed a Shapiro–Wilk test to evaluate data distribution. We used Chi-squared and Fischer’s exact test for proportion, two-sided *t*-tests, or Mann–Whitney tests for continuous variable. We also performed a multinomial logistic regression in order to evaluate the associations between clinical parameters with MSA/MAA positivity. The analysis was performed with the inclusion of all parameters proven to be associated with positivity for MSA and/or MAA.

Data were presented in proportion or in mean (±standard deviation, SD), *p* value and 95% confidence interval (95CI). Values of *p* < 0.05 were considered statistically significant.

Missing data were less than 1%, and mostly pertained to pulmonary function tests, due to the inability of some patients to carry out the exam. Where data were missing, Table 1 specifies how many patients were studied for the single item.

## 3. Results

During the study period, 518 patients with ILD were evaluated at our pulmonology clinic. The number of patients excluded were 351. Sixty-three patients did not complete the diagnostic assessment, four patients did not give their consent, and 284 did not show features of IIM or M+. On this latter group of patients, we performed only the common ENA profile and did not find any positivity for anti-Jo1. The remaining 167 patients with suspected myositis (based on the criteria reported above) were enrolled and divided into two groups according to the presence of M+. Sixty-three patients were included in the M+ILD+ cohort and 104 in the M-ILD+ cohort. In the rheumatological setting the presence of M+ is quite common. For the purposes of the study, we considered 544 myalgia patients, of whom 324 were excluded due to the presence of clinical signs raising suspicion of an underlying autoimmune condition, 38 due to an incomplete diagnostic assessment and 8 due to lack of consent. Finally, we enrolled 174 M+ patients without ILD (M+ILD-).

ILD patients had similar ages (M+ILD+ 62.9 ± 10.6 years, M-ILD+ 64.3 ± 10.5 years, *p* = not significant, n.s.). Gender was significantly different between the groups. In the M+ILD+ group 90.5% were females, whereas in the M-ILD+ group only 52.9% were females (*p* = < 0.0001, X^2^ = 25, 95CI 24.1–48.4). The M+ILD- group was composed of 92% females. The proportion was similar to that reported in M+ILD+ group, but higher than the M-ILD+ patients (M-ILD+ *p* = < 0.0001, X^2^ = 56.6, 95CI 28.5–49.2). The M+ILD- patients had a mean age of 57.7 ± 13.9 years, significantly younger than the M+ILD+ patients (*p* = 0.002 95CI −9.5–−3.7) and M-ILD+ patients (*p* =< 0.0001 95CI −9.5–−3.7).

The clinical presentation at the first ILD assessment is reported in Table 1.

At the first assessment, all ILD patients were aged ≥18 years. Spontaneous myalgia was reported by 43 ILD patients (25.7%), and 31 (18.6% of the whole cohort, 49.2% of myalgia patients) were positive for our test. Considering the specific features of IIM, our ILD patients showed dysphagia in 40.7% of cases, proximal weakness in 23.3%, and typical skin rashes (Gottron’s sign/papules or heliotrope rash) in 6.6%. Regarding muscle enzymes, AST and ALT were normal, but increased levels of CPK and/or LDH were noted in 25.7% of patients. At least one specific feature was present in 59.3% of patients enrolled (40.1% excluding dysphagia). As reported in Table 1, stratifying according to the presence of myalgia, no differences were noted.

MSAs were positive in 38.1% of M+ILD+ patients, 20.2% of M-ILD+ and 8% of M+ILD- patients. MAAs were positive in 50.8% of M+ILD+ patients, 36.5% of M-ILD+ and 8% of M+ILD- patients. M+ILD+ patients were positive for at least one MSA/MAA in 68.3% of cases, M-ILD+ in 48.1% and M+ILD- in 17.2%.

Figure 2 reports the difference between the groups.

Regarding MSA/MAA, we did not find differences in prevalence between M+ILD+ and M-ILD+ patients. M+ILD- patients showed a lower prevalence of anti-Jo1 than both of the other groups (vs. M+ILD+ *p* = 0.0001, vs. M-ILD+ *p* = 0.0005), and similar results were noted with anti-Ro52Kd (vs. both of the other groups *p* = < 0.0001). M+ILD- also had a lower prevalence of anti-PL7 (*p* = 0.006), anti-Pm/scl (*p* = 0.007) and anti-RNP (*p* = 0.04) compared with M+ILD+ patients. The prevalence of each autoantibody tested in the three groups is reported in Appendix A. Multiple seropositivity for MSA/MAA was found in 20.6% of M+ILD+, 11.5% of M-ILD+ and 4% of M+ILD- patients. Considering the association between MSAs, the proportion was 3.2%, 1.9% and 2.9%, respectively. We report the associations in Appendix A.

Considering all ILD patients (M+ and M−, 167 patients), positivity for MSAs was associated with myalgia (X^2^ 6.4, *p* = 0.01), dysphagia (X^2^ 4.1, *p* = 0.04), proximal weakness (X^2^ 7.1, *p* = 0.007), and typical skin rashes (Gottron’s papules/sign, heliotrope rash, X^2^ 8.1, *p* = 0.005). Positivity for MAAs was associated with the presence of puffy fingers (X^2^ 9.6, *p* = 0.002), RP (X^2^ 7.7, *p* = 0.005), ANA positivity (X^2^ 16.9 *p* < 0.0001), and the presence of a speckled ANA pattern (X^2^ 5.2, *p* = 0.02). Positivity for at least one MSA/MAA was associated with the presence of myalgia (X^2^ 6.5, *p* = 0.01), telangiectasia (X^2^ 4, *p* = 0.04), puffy fingers (X^2^ 6.3, *p* = 0.01), ANA positivity (X^2^ 10.9 *p* = 0.001), and RP (X^2^ 8.2, *p* = 0.004). The associations between the different items evaluated in the study are reported in Appendix A.

The clinical parameters associated with positivity for MSA and MSA/MAAs were included in a multiple logistic regression (Table 2). The presence of myalgia confirmed its association with seropositivity.

We also evaluated the association of myalgia with MSA/MAA positivity in subjects with a lower risk of having IIM at baseline, therefore excluding those patients with typical skin rashes, increased levels of CPK and LDH, or proximal weakness. Additionally, in this case, the association was statistically significant (X^2^ 5.5 *p* = 0.01). Myalgia in the entire cohort showed a sensitivity of 46% and specificity of 73%, while in patients at a lower risk of underlying IIM, sensitivity was 48% and specificity 71.2%.

Finally, we evaluated the diagnoses made by the multidisciplinary team in ILD patients (M+ILD+ and M-ILD+). The M+ILD+ group received a significantly higher proportion of diagnoses of IIMs than M-ILD+ (39.7% vs. 23.1%, *p* = 0.02, X^2^ = 5.2, 95CI 2.3–30.8). In two cases, the diagnosis of IIM was made without any autoantibody positivity, using magnetic resonance imaging (MRI) and/or histologic exams. In ILD patients, we performed 15 MRI and 12 muscle biopsies looking for IIM, and all of them proved to be positive. ILD patients displayed increased muscle enzymes in 45 cases, in one case myoglobin and in another aldolase (enzymes not included in the IIM classification criteria). The final diagnoses in these patients was IPAF and overlap SSc-PM, respectively. None of the 13 patients with increased levels of LDH had a final diagnosis of IIM, in contrast with the 30 patients with increased levels of CPK (27 with CPK alone and 3 combined with LDH), where a final diagnosis of IIM was made in 60% of cases. Appendix A reports the diagnoses in detail. M+ILD+ also showed a higher proportion of definite autoimmune rheumatic diseases (ARDs) than M-ILD+ patients (*p* = 0.005 X^2^ = 7.9, 95CI 5.6–32.8), while the proportion of IPAF patients was similar between the two groups. The proportion of specific autoimmune conditions in the two ILD cohorts is reported in Figure 3, while the proportion of different diagnoses is reported in Appendix A.

In M+ILD-, we found four patients with primary Sjögren’s syndrome (pSS) and one with rheumatoid arthritis, systemic lupus erythematosus and AS. Another 26 patients were classified as UCTD, while the other patients showed no inflammatory conditions. All patients positive for MSA/MAA underwent HRCT, but only the patient finally classified as AS showed ILD with an asymptomatic nonspecific interstitial pneumonia (NSIP) pattern.

## 4. Discussion

ILD and cancer are the most common mortality factors in IIMs making the diagnosis of IIMs in general and their association with ILD particularly important [34]. Considering the frequent progressive-fibrosing phenotype of ILDs, the recognition of the association between IIMs and ILD could allow early treatment of the respiratory involvement, together with appropriate screening for malignancy. An early recognition and subsequent immunosuppressive treatment could also prevent the appearance or worsening of other clinical manifestations of the disease, which are very common in the first year after diagnosis [3].

However, the diagnosis is still very difficult in a non-rheumatological setting. Frequently, IIM-ILD patients, even with severe lung involvement, show normal or mildly increased serum level of muscle enzymes both at diagnosis and during follow-up [3]. IIMs is the autoimmune condition that is most frequently associated with OP-ILD, although all HRCT patterns may be present in ILD-IIM [16]. These patients may also have positive NVC, not necessarily associated with the presence of RP, thus making them difficult to recognize [21]. Typical skin rashes are specific, although rare, and the recognition can be difficult for physicians who are not specifically trained.

In this study, we investigated whether, in a clinical setting where the approach to ILDs and autoimmune diseases is multidisciplinary, myalgia could be evaluated not only as an unspecific symptom, but as a possible predictive factor of positivity for MSA/MAAs.

The recognition of these autoantibodies is very helpful for the management of ILD patients. Despite the fact that only ant-Jo1 is included in classification criteria for IIM [9], other MSA/MAAs can suggest the presence of AS or at least IPAF [7,8]. MSA/MAAs other than anti-Jo1 proved to be closely associated with severe ILD as the possible main feature, or even sole clinical involvement of a CTD [3,18].

In order to maximize the pre-test probability of finding positive MSA/MAA, we enrolled ILD patients with suspected IIMs based on several risk factors, as well as a control group of myalgia patients without ILD. Myalgia is currently not included in ARD classification criteria [7,8,9]. However, myalgia is mainly reported by seropositive IIM patients with a prevalent involvement of the muscular fascia, with the same patients facing a higher risk of developing ILD [6]. Based on this, we suppose that in a respiratory setting, the clinical evaluation of muscular fascia could predict positivity for MSA/MAAs. We chose to study seven different points (a total of 14) in which proximal muscles had a superficial insertion and were easy to evaluate.

Some of the points evaluated in this study were used in the past for the diagnosis of fibromyalgia, but were excluded in 2010 from the classification criteria of the diagnosis of this condition due to their variability, poor sensitivity to change during follow-up, and excessive sensitivity in the recruitment of women [35]. Fibromyalgia is a very common non-inflammatory rheumatic condition, characterized by widespread musculoskeletal pain, fatigue, sleep problems, and mood issues. Until 2010, the diagnosis was made with at least 11/18 positive tender points [35]. The aim of the study was to evaluate the clinical assessment of myalgia as a possible sensitive method for the recognition of MSA/MAA positivity, looking for an oligo-amyopathic form of IIM. To avoid possible overlap with fibromyalgia (the diagnosis of fibromyalgia is outside the aim of this study) we evaluated only entheseal points of the proximal muscles, and, looking for the recognition of oligo-amyopathic form of IIM, we decided to enroll patients with at least five painful points.

Patients with ILD were older than M+ILD- patients, and displayed similar clinical features, disease severity, and HRCT presentation, regardless of the presence of myalgia. The sole difference at baseline was a higher proportion of women in the M+ILD+ cohort. These data were predictable, considering that myalgia is more common in women [35], however it should be considered that women also have a higher prevalence of CTD in general and IIM in particular [36]. M+ILD+ patients showed a higher prevalence of MSAs and MAAs than the other two groups. Seropositivity in this group oriented us towards the diagnosis of ARDs, and in particular IIMs, in a proportion significantly higher than reported in the M-ILD+ group.

Considering all the ILD patients together, the increased levels of muscle enzymes and dysphagia were not associated with seropositivity for MSA/MAA in ILD patients. As already mentioned, MSAs other than anti-Jo1 (e.g., anti-PL7 or anti-PL12) are mainly associated with severe ILD and very mild myositis [3,18]. The increased level of LDH in our cohort could better reflect lung damage than myositis [11], and dysphagia is very common in all ILDs, particularly IPF [12]. Other well-established risk factors for IIM, such as proximal weakness and typical skin rashes, proved to be associated only with MSA positivity, whereas myalgia was associated with both MSA and MSA/MAA positivity, also in multivariate models, including the above-mentioned clinical features. Therefore, myalgia in ILD patients with suspected myositis could have an additive value for the recognition of MSA/MAA positivity. Moreover, we were surprised to note that in our study, M+ILD- patients were seropositive for at least one MSA/MAA in 17.2% of cases. Only one seropositive patient had lung involvement (a patient with AS), but in the entire cohort, we made a specific diagnosis of ARDs in 4% (7 patients), of which 2.3% were composed of patients with pSS. These patients did not report a significant sicca syndrome; however, after the recognition of seropositivity for SSA, the tests confirmed the diagnosis according to the specific criteria [26]. These data are surprising, considering that the prevalence in the general population of pSS is about 0.25% [37], but myalgia is reported in up to 31% of pSS patients, and myalgia was previously suggested as a potential risk factor for the development of pSS [38,39]. We can suppose that in our cases, myalgia allowed a diagnosis of pSS before the sicca syndrome became symptomatic.

Although we cannot exclude the possibility of some false positive results, patients seropositive for MSA/MAAs were classified as UCTD. Limited data are available regarding the prevalence of MSA/MAA in healthy controls. A single study reported a prevalence of about 50% of these autoantibodies in IIMs patients (n = 146), compared with 3% in other rheumatic diseases (n = 200) and 0% in healthy subjects (n = 40) [40]. Another study using IB found 48% of MSA/MAA positivity in IIM patients (n = 110) and 5% in healthy controls (n = 60) [41]. In our cohort of M+ILD- patients in whom the sole clinical feature was myalgia, the prevalence was significantly higher (17.2%). Despite the relatively common presence of MSA/MAA in our M+ILD- cohort, we found only one case of asymptomatic ILD. The main difference between our ILD patients and the cohort of myalgia patients without ILD was the lower proportion of MSA/MAA tightly associated with ILD: anti-Ro52Kd, Jo1, anti-Pm/scl and PL7 [42,43]. Common autoantibodies, such as ANA and ENA, were negative, or present in a proportion usually found in healthy subjects [44].

The proportion of MSA/MAA positivity in our myalgia patients without ILD (17.2%) was similar to that reported by a recent, retrospective work regarding the prevalence of these autoantibodies in patients with suspected IIM (17.7%) [45]. These data could also support the possibility of an association between myalgia and MSA/MAA positivity. In view of this, a possible pathogenic role of MSA/MAA in myalgia patients (or a possible, incomplete form of IIM) could be hypothesized.

In our opinion, these data underline the importance of follow-up in myalgia patients, mainly for those who present clinical or serological features suggestive of, but not sufficient for, the diagnosis of autoimmune disorders.

Finally, we should consider the possible presence of false positivity in our cohort. We used IB for the identification of MSA/MAAs, finding multiple positivity in some cases. The gold standard for the recognition of these autoantibodies remains immunoprecipitation (IP); however, it is very difficult to use in clinical practice. Even using IP, the association between MSA/MAA is quite common, reaching about 50%, whereas MSAs are considered mutually exclusive and their association very uncommon (0.3%) [44]. An interesting paper from 2016 by Cavazzana et al., compared the performance of IB and IP in the recognition of MSA/MAAs, reporting an association between MSA in 17% when evaluated in IB, and 0% with IP [46]. Another study from 2019 evaluated IB vs. IP using a similar kit, and found a higher concordance [41]. In our study, we used a kit from the same manufacturer, and multiple positivity for MSA was found in a proportion significantly lower to that reported in 2016 (only 3.2% in M+ILD+ patients, 1.9% in M-ILD+ patients and 2.9% in M+ILD- patients).

This study has some limits. Firstly, our M+ILD+ patients showed other clinical, serological, and radiological features suggestive of IIM; therefore, the actual role of this sign should be evaluated together with other clinical signs that might be suggestive for IIMs, and the role of our myalgia points alone in ILD patients should be addressed. Secondly, our hypothesis of a possible inflammatory involvement of the muscular fascia needs to be demonstrated with instrumental examinations. Moreover, one of the major concerns regarding the evaluation of myalgia (higher sensitivity in the female population) seems to be confirmed in our study, despite the fact that IIM is, regardless, more frequent in female subjects [36]. We used IB for the recognition of MSA/MAAs, while the gold standard for their assessment remains IP, although it is very difficult to use in clinical practice. Finally, the diagnostic value of myalgia in ILD patients needs to be confirmed and validated by an external cohort, looking also at whether the cut-off of 5 points is appropriate.

However, we believe that this study has some merits. The evaluation of myalgia is simple with a rapid learning curve, safe, timesaving, and totally free. Currently, objective respiratory exams are aimed at finding signs of ILD, but do not always consider additional elements suggestive of the possible etiologic cause of an underlying ILD. The same inflammatory arthritis necessitates appropriate training in its correct identification. This is especially true for rare skin rashes, such as heliotrope rash or Gottron’s sign. Myalgia evaluation could also be a simple and useful tool for non-rheumatologists.

## 5. Conclusions

The finding of myalgia in proximity to the entheses in patients with ILD and suspected myositis could identify patients with higher likelihood of positivity for MSAs and/or MAAs, and this positivity could drive a specific diagnosis of ARD, and in particular, IIMs. In these patients, a second-line autoimmune serological assessment should be performed, looking for MSA/MAAs. There is a need for validated guidelines suggesting when MSA/MAA tests should be performed; however, physicians could consider the presence of myalgia, evaluated with our method, as a possible suggestive element.

## Figures and Tables

**Figure 1 diagnostics-12-01139-f001:**
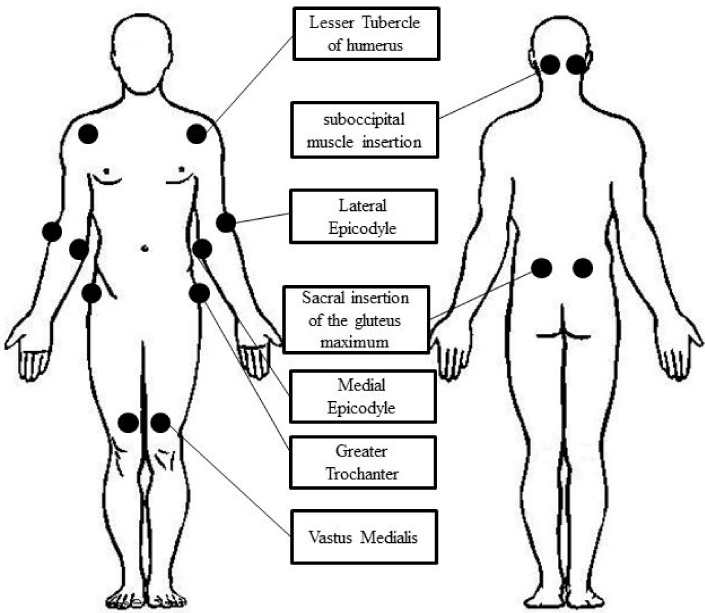
Graphical Representation of the point evaluated for the assessment of Myalgia.

**Figure 2 diagnostics-12-01139-f002:**
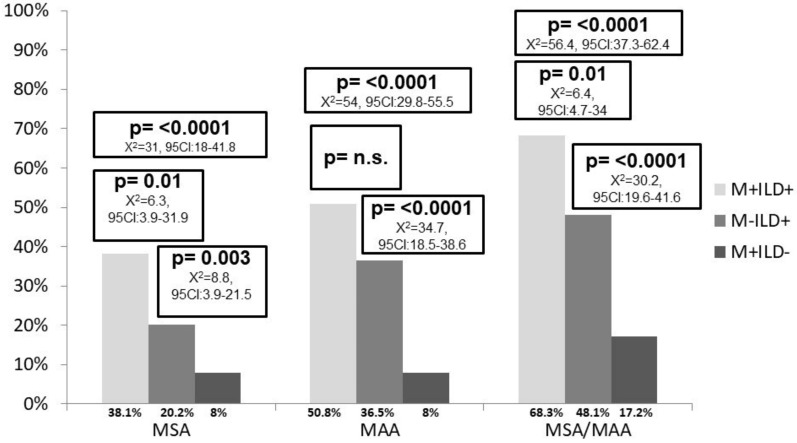
Proportion of MSA/MAA positivity in the three cohorts studied. Figure 2 legend: ILD: interstitial lung Disease; M: myalgia; MAA: myositis-associated antibodies; MSA: myositis-specific antibodies.

**Figure 3 diagnostics-12-01139-f003:**
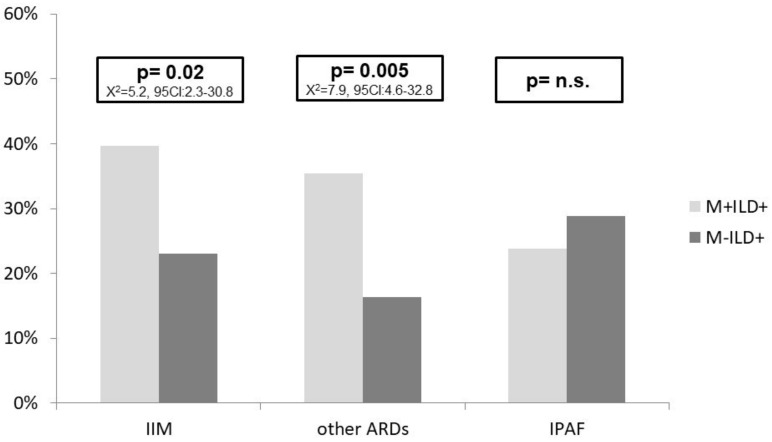
Diagnosis in ILD patients. Figure 3 Legend: ARD: autoimmune rheumatic disease; IIM: idiopathic inflammatory myopathies; ILD: interstitial lung disease; IPAF: interstitial pneumonia with autoimmune features; M: myalgia. IIM: in this group we included dermatomyositis (DM), polymyositis (PM), antisynthetase syndrome (AS), overlap syndrome (OS) including myositis. ARD: in this group we included patients with specific diagnosis of rheumatoid arthritis (RA), systemic lupus erythematosus (SLE), mixed connective tissue disease (MCTD), primary Sjögren’s syndrome (pSS), systemic sclerosis (SSc), OL condition not including myositis, vasculitis.

**Table 1 diagnostics-12-01139-t001:** Clinical Presentation of ILD patients.

Item	M+ILD+ (*n* = 63)	M-ILD+ (*n* = 104)	*p*
Age	62.9 ± 10.6	64.3 ± 10.5	n.s.
Female	90.5%	52.9%	**<0.0001** **X^2^ = 25** **95CI:24.1–48.4**
Arthritis/PMR	22.2%	34.6%	n.s.
RP	49.2%	30.8%	**0.02** **X^2^= 5.6** **95CI: 3.2–32.9**
Sclerodattilia/puffy fingers	17.5%	12.5%	n.s.
Telangiectasias	14.3%	8.7%	n.s.
Skin rashes #	17.5%	9.6%	n.s.
Gottron’s papules/sign, heliotrope rash	11.1%	3.8%	n.s.
Unexplained fever	30.1%	25%	n.s.
Sicca syndrome	34.9%	25%	n.s.
All increased seric muscle enzymes	22.2%	29.8%	n.s.
Increased CPK and/or LDH	19%	29.8%	n.s.
Dysphagia	44.4%	38.5%	n.s.
NVC+	20.6%	21.2%	n.s.
Bushy capillaries in NVC	27%	37.5%	n.s.
Exocrine gland functional tests §	19%	16.3%	n.s.
History of cancer	14.3%	6.7%	n.s.
Accompanying features of IIMs *	3.9 ± 2	3.6 ± 1.9	n.s.
At least 1 feature specific for IIM °	44.4%	38.5%	n.s.
** *Respiratory features* **			
Dispnoea	92.1%	90.4%	n.s.
Cough	49.2%	48.1%	n.s.
FVC%	87 ± 25.6	86 ± 23.2	n.s.
DLCO%	65 ± 20.4	63.7 ± 20.1	n.s.
Oxygen support	36.5%	34.6%	n.s.
** *Radiologic pattern of ILD* **			
NSIP	61.9%	48.1%	n.s.
OP	11.1%	17.3%	n.s.
UIP-like	27%	37.5%	n.s.
Combined	6.4%	6.7%	n.s.
Indeterminate	4.7%	3.8%	n.s.

Legend: CPK: creatine phosphokinase; DLCO%: diffusing lung capacity for carbon monoxide in proportion of the predicted; FVC%: forced vital capacity in proportion of the predicted; ILD: interstitial lung disease; LDH: lactic dehydrogenase; M: myalgia NSIP: nonspecific interstitial pneumonia; NVC: nailfold videocapillaroscopy; OP: organizing pneumonia; PMR: polymyalgia rheumatica; UIP: usual interstitial pneumonia; combined: among M+ILD+ patients, 4 subjects showed a combination of NSIP+OP pattern, while in the M-ILD+ group 8 patients had NSIP+OP pattern, and a single patient NSIP+UIP; § = carried out on in 36 patients in M+ILD+ and 55 patients in M-ILD+; * = defined as reported in the method section; #: Gottron’s sign/papules, or heliotrope rash; ° = at least 1 feature from: dysphagia, proximal weakness, increased level of CPK or LDH, presence of Gottron’s sign/papules, or heliotrope rash; PFT were performed in 55 M+ILD+ and 99 M-ILD+ patients.

**Table 2 diagnostics-12-01139-t002:** Association of clinical features with positivity for MSA/MAA.

Dependent Variable: Positivity for MSA		
Item	*p*	OR (95%CI)
Myalgia	0.02	2.47 (1.2–5.3)
Typical skin rashes *	0.06	3.6 (0.96–14)
Proximal weakness	0.03	2.6 (1.1–6)
Dysphagia	0.31	1.5 (0.7–3.2)
**Dependent Variable: Positivity for MSA/MAA**		
Item	*p*	OR (95%CI)
Myalgia	0.04	2.1 (1–4.1)
Raynaud’s phenomenon	0.04	2.1 (1–4.4)
Puffy fingers	0.22	2.2 (0.6–8.4)
Telangiectasia	0.93	0.93 (0.2–4.4)
Typical skin rashes *	0.35	2.2 (0.4–11.9)
Proximal weakness	0.18	1.7 (0.8–4)
Dysphagia	0.29	1.5 (0.7–3)

Legend: 95CI: 95% confidence interval; MAA: myositis-associated antibody; MSA: myositis-specific antibody; OR: odds ratio; *: heliotrope rash, Gottron’s papules/sign.

## Data Availability

The data presented in this study are available on request from the corresponding author. The data are not publicly available due to ethical reasons.

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
