# Peer review of "A New Method for the Assessment of Myalgia in Interstitial Lung Disease: Association with Positivity for Myositis-Specific and Myositis-Associated Antibodies"

_diagnostics, 2022, doi:10.3390/diagnostics12051139_

Round 1
Reviewer 1 Report
Comments to the Author
The authors suggested that myalgia in ILD patients with suspected myositis is associated with MSA/MAA positivity and a diagnosis of IIM in the present study. This study has clinical significance. However, the resolution for the following query is needed for the acceptance to the BMC Pulmonary Medicine
Major comments
- This study evaluated the significance of analyzing the clinical evaluation of the muscular fascia. I would like to know the accuracy of the evaluation of muscle symptoms by this method. Please reveal any data of concordance between the clinical fascial pain assessment and the myositis assessment by muscle biopsy or EMG in this study dataset or previous reports.
- As the author shows, the gold standard for MSA / MAA measurement is IP, and the problem is that the concordance with the result analyzing autoimmune antibodies measured by IP and IB differs for each antibody. The analysis results of some cases are acceptable, so could you Please show the concordance between the antibody evaluations measured by IP and IB in this dataset, if the analysis is possible.
- Please explain why you are not referring to the latest guidelines even though you are diagnosing a group of cases that started enrollment from 2021 for diagnosis of IIP.
Author Response
Comments to the Author
The authors suggested that myalgia in ILD patients with suspected myositis is associated with MSA/MAA positivity and a diagnosis of IIM in the present study. This study has clinical significance. However, the resolution for the following query is needed for the acceptance to the BMC Pulmonary Medicine
Major comments
- This study evaluated the significance of analyzing the clinical evaluation of the muscular fascia. I would like to know the accuracy of the evaluation of muscle symptoms by this method. Please reveal any data of concordance between the clinical fascial pain assessment and the myositis assessment by muscle biopsy or EMG in this study dataset or previous reports.
REPLY: Thank you for your comment. The aim of the study is mainly to evaluate myalgia as a possible predictor or MSA/MAA positivity in ILD patients rather than the presence of underlying myositis. As reported in the study limitations, our hypothesis of a possible inflammatory involvement of the muscular fascia needs to be demonstrated with instrumental examinations. Muscle biopsy and EMG, although performed on some patients, were not systematically assessed, because they are not mandatory for the diagnosis of IIM, PM, DM or AS according to the current criteria (Lundberg IE 2017, Bohan & Peter 1975, Connors 2010 respectively). To the best of our knowledge, the sole study that directly associates myalgia and histologic findings in Idiopathic Inflammatory Myopathies was carried out by Noda K et al (ref.6). The authors proved that myalgia is mainly associated with fasciitis rather than myositis (explained in lines 67-73). Other papers proved that in IIM patients, myalgia is more common in patients with MSA positivity. This latter point has been added in the same paragraph (ref.5). A direct correlation between myalgia and EMG in IIM patients was not found, but EMG is useful only to support IIM diagnosis: EMG findings in IIM are similar to other infectious, toxic or metabolic myopathies.
- As the author shows, the gold standard for MSA / MAA measurement is IP, and the problem is that the concordance with the result analyzing autoimmune antibodies measured by IP and IB differs for each antibody. The analysis results of some cases are acceptable, so could you Please show the concordance between the antibody evaluations measured by IP and IB in this dataset, if the analysis is possible.
REPLY: Unfortunately, we only evaluated these patients with IB. We discussed this point as a limit of the study in the discussion section, however, in the same paragraph, we specified that we used the same kit as reference 38, obtaining similar results, which gives reassurance about the test’s reliability (lines 433-440). We also believe that IP is clearly the gold standard for the evaluation of MSA/MAA positivity, however clinical practice is largely based on IB, and a clinical test able to predict positivity for MSA/MAA could support the physician in diagnostic assessment.
- Please explain why you are not referring to the latest guidelines even though you are diagnosing a group of cases that started enrollment from 2021 for diagnosis of IIP.
REPLY: Actually, the cohort we took into consideration is composed of ILD patients with suspected myositis, and the majority of these guidelines are not applicable, mainly due to the presence of a HRCT pattern suggesting an alternative diagnosis (according to the same guidelines). However, clearly in some cases the diagnosis of IPF was made, and we are very grateful for your careful checking of our manuscript. We realized that, probably confused by the same first author, we reportet 2011 guidelines rather than 2018 ones. We have now fixed this error.
Reviewer 2 Report
This study evaluates the value of a method of assessing myalgia in a selected population of ILD patients at high risk of IIM as a screening tool for patients with IIM or MSA/MAAs.
Although interesting, there are several limitations that severely restrict the clinical utility of this method in my opinion:
- The cohort is selected as being at high risk of IIM, so it is in any case a group in which IIM will be looked for using antibodies test, MRI, PET CT or biopsy, the value of myalgia in an unselected cohort would be more relevant for clinical practice and to guide clinicians.
- It is not known whether these patients had spontaneous myalgia so we can hypothesize that spontaneous proximal myalgia may be suficient to search for IMM and the added value of the described method remains in this case controversial.
-This test appears to me as a rheumatologist as very unspecific , as it can be positive in several fibromyalgic patients: a test for specificity and sensibility for this clinical method in an unselected group of ILD patients will be of interest to better discriminate the value of this method in clinical setting
-The diagnostic method of IIM in this study is not well described: MRI, PET CT and/or biopsy should be used , the multidisciplinary diagnosis remains unclear and prevents any correlation between myalgia and pathological/MRI fundings.
-Muscular enzymes values should be presented and compared between patients with and without IIMs , as a higher level may be indicative of IIM; the lack of presentation of these data prevents any analysis and correlation
- The multivariate model should be presented.
- There are some small spelling mistakes with words in Italian
Author Response
REWIEWER 2
This study evaluates the value of a method of assessing myalgia in a selected population of ILD patients at high risk of IIM as a screening tool for patients with IIM or MSA/MAAs.
Although interesting, there are several limitations that severely restrict the clinical utility of this method in my opinion:
- The cohort is selected as being at high risk of IIM, so it is in any case a group in which IIM will be looked for using antibodies test, MRI, PET CT or biopsy, the value of myalgia in an unselected cohort would be more relevant for clinical practice and to guide clinicians.
REPLY
Thank you for your comment. As currently described, the study clearly raises this objection, and need to be clarified.
The objective of this study is to evaluate the role of myalgia as possible predictor of positivity for any MSA/MAA in a cohort of ILD patients suspected of underlying IIM.
Based on the current ACR/EULAR criteria (Lundberg IE 2017, cited in the text), the suspicion of IIM can be sustained based on the age of onset (18-40 years and >40y), proximal weakness, presence of typical skin rashes (heliotrope rash, Gottron’s sign/papules), increased serum level of muscular enzymes (AST, ALT, CPK, LDH) and dysphagia. However, our definition of “suspected IIM” is significantly wider, including other signs able to support the diagnosis but not included in any classification criteria (e.g. nailfold videocapillaroscopic findings, ANA patterns, mechanic’s hands, hiker’s feet, Organizing Pneumonia in HRCT, unexplained fever, puffy fingers, Raynaud’s phenomenon, and several others assessed in the Methods section). Therefore, we believe that our population is composed of a cohort of ILD patients with only a slightly increased risk of IIM. This point is now specified in lines 102-119.
Excluding the age of onset (ILD before 18 years is very uncommon and not present in our cohort), we can consider at actual risk of IIM those patients with at least 1 of the other signs. In our population proximal weakness was reported by 23.3% of patients, typical skin rashes by 6.6% of patients, increased muscle enzymes by 25.7% and dysphagia by 40.7%. At least one of the first three signs is present in 40.1% of patients and including dysphagia (a well-recognized risk factor for all ILDs, and above all Idiopathic Pulmonary Fibrosis), 59.3%. We reported these data in lines 227-234 and in table 1.
In our ILD cohort, dysphagia, typical skin rashes, and proximal weakness (but not the increased level of muscle enzymes) were associated only with positivity for MSA. Myalgia was associated with both MSA and MSA/MAA positivity, and confirmed its possible predictive role also in multiple logistic regression including the above mentioned parameters, well-recognized risk factors for IIM. Therefore, we can hypothesize that myalgia could represent an added value in the diagnostic evaluation of ILD patients. These data are now reported in lines 255-268 and table 2.
It is not known whether these patients had spontaneous myalgia so we can hypothesize that spontaneous proximal myalgia may be suficient to search for IMM and the added value of the described method remains in this case controversial.
REPLY
Thank you for your comment. Spontaneous myalgia was reported by 43 ILD patients (25.7%) and 31 (18.6% of the whole cohort, 49.2% of myalgia patients) were found to be positive with our test. No association was found between spontaneous proximal myalgia and MSA/MAA positivity. You can find these data at lines 227-229.
3.This test appears to me as a rheumatologist as very unspecific , as it can be positive in several fibromyalgic patients: a test for specificity and sensibility for this clinical method in an unselected group of ILD patients will be of interest to better discriminate the value of this method in clinical setting
REPLY: We totally agree with your opinion. The test is frequently positive in fibromyalgic patients. As reported in the text, we enrolled 174 myalgia patients without any clinical suspicion of autoimmune disease, and we found seropositivity for these autoantibodies in 17.2%, making a specific diagnosis of ARDs in 4%, and in particular of primary Sjögren’s syndrome in 2.3%. The data is quite surprising: considering that the prevalence of pSS in the general population is reported at 0.25%, in our rheumatologic setting it was about 10 times greater. We already assessed this point in lines 394-404. As fibromyalgia is diagnosed by exclusion, we considered FMG all those myalgia patients proved to be seronegative for any antibody. As already assessed in lines 405-423, we cannot exclude possible cases of false positivity considering the use of IB instead of IP in the evaluation of MSA/MAA, however we used the same kit that demonstrated great reliability in previous papers. Moreover, the proportion of our M+ILD- patients positive for MSA/MAA is quite similar to that reported in unselected patients with suspected IIM (reference 45).
The value of myalgia in predicting MSA/MAA positivity in ILD patients needs to be demonstrated also in unselected ILD patients in any case. This is clearly reported as a limit of the study at lines 444-446. However, we tried to evaluate the diagnostic value of myalgia in patients at lower risk of having underlying IIM, excluding patients with a typical skin rash, increased levels of CPK or LDH and proximal weakness (the items with the greatest diagnostic weight in IIM criteria). Also in this case, myalgia proved to be statistically significant (you can find these data at lines 271-276).
4.The diagnostic method of IIM in this study is not well described: MRI, PET CT and/or biopsy should be used , the multidisciplinary diagnosis remains unclear and prevents any correlation between myalgia and pathological/MRI fundings.
REPLY: As declared in the manuscript, the aim of the study is to clinically evaluate myalgia in muscular fascia as a possible predictor of positivity for MSA/MAA in ILD patients, and not strictly the diagnosis of IIM. ILD is commonly the first, the main or even the sole clinical manifestation of a CTD, mainly for pSS and IIM, which are conditions closely associated with MSA/MAA positivity (ref 3 and 42). However, the recognition of MSA/MAA positivity can be useful to classify patients as at risk of developing a definite form of CTD (IPAF classification, ref 7), to stratify the risk of development of CTD in patients not classifiable as IPAF (e.g. sicca patients, considering that anti-Ro52 are not always included in common ENA commercial kits), the risk of progressive fibrosing ILD (e.g. ILD patients positive for PL7, PL12, Ro52) or even acute exacerbation of ILD and cancer (MDA5 and Tif1gamma respectively). However, we agree with you that, considering the aim of the study, the actual diagnosis of IIM is of interest. As assessed in the manuscript, the diagnosis of IIM was made according to the criteria of Lundberg IE et al 2017, PM and DM using Bohan & Peter criteria 1975 and, for AS, Connors criteria 2010 (lines 175-177). None of these criteria required positivity for MRI, PET CT and/or biopsy, however these exams were performed when deemed useful for correct diagnostic assessment (lines 170-172). Considering only ILD patients, we performed 15 MRIs and 12 muscle biopsies, all positive for myopathies (now reported in lines 287-295). However, as reported in the study limitations, our hypothesis of a possible inflammatory involvement of the muscular fascia still needs to be specifically demonstrated with instrumental examinations.
- Muscular enzymes values should be presented and compared between patients with and without IIMs, as a higher level may be indicative of IIM; the lack of presentation of these data prevents any analysis and correlation
REPLY: A direct comparison was not made because most of our patients showed normal levels of muscular enzymes. An increased level was found in 25.7% of ILD patients: 1 patient with increased levels of myoglobin, 1 with aldolase, 13 with LDH, 24 with CPK and 3 with LDH and CPK. The patient with increased levels of myoglobin was classified as IPAF and the patient with increased aldolase as overlap DM-SSc.
We had also 13 patients with increased levels of LDH: in this group the patients were classified as idiopathic NSIP (5), IPF (4) and IPAF (4). The lack of specificity of LDH in the recognition of IIM patients could be explained by the fact that lung damage increases serum levels of LDH also without muscle involvement (explained in lines 105-107). There were 27 patients with increased levels of CPK. The diagnoses were: AS (9 patients), RA (2 patients) PM (3 patients) Overlap PM-SSc (2), DM-SSc (1), SLE-PM (1), IPAF (4), IPF (2). The 3 patients with combined increased levels of CPK and LDH were classified as AS, PM and pSS. These data are now reported at lines 292-295 and in table S4 in detail.
- The multivariate model should be presented.
REPLY: Thank you for your suggestion. We have now presented our multivariate models in lines 257-269, table 2, and table S3
- There are some small spelling mistakes with words in Italian
REPLY: Thank you, we reviewed the manuscript with a native English, we hope that the manuscript is now improved.
Round 2
Reviewer 1 Report
Authors have revised this paper correctly.
Author Response
we are very grateful for your suggestions.
Reviewer 2 Report
All my comments have been adequately taken into account. The manuscript has been largely improved.
Minor comment: please add 95% CI in Table 2
Author Response
we are very grateful for your suggestions. The 95%CI was added in table 2.